# AmAtg2B-Mediated Lipophagy Regulates Lipolysis of Pupae in *Apis mellifera*

**DOI:** 10.3390/ijms24032096

**Published:** 2023-01-20

**Authors:** Wen-Feng Chen, Hong-Fang Wang, Ying Wang, Zhen-Guo Liu, Bao-Hua Xu

**Affiliations:** College of Animal Science and Technology, Shandong Agricultural University, Tai’an 271018, China

**Keywords:** honey bees, development, fat bodies, autophagy, LDs

## Abstract

Lipophagy plays an important role in regulating lipid metabolism in mammals. The exact function of autophagy-related protein 2 (Atg2) has been investigated in mammals, but research on the existence and functions of Atg2 in *Apis mellifera* (AmAtg2) is still limited. Here, autophagy occurred in honeybee pupae, which targeted lipid droplets (LDs) in fat body, namely lipophagy, which was verified by co-localization of LDs with microtubule-associated protein 1A/1B light chain 3 beta (LC3). Moreover, AmAtg2 homolog B (AmAtg2B) was expressed specifically in pupal fat body, which indicated that AmAtg2B might have special function in fat body. Further, AmAtg2B antibody neutralization and AmAtg2B knock-down were undertaken to verify the functions in pupae. Results showed that low expression of AmAtg2B at the protein and transcriptional levels led to lipophagy inhibition, which down-regulated the expression levels of proteins and genes related to lipolysis. Altogether, results in this study systematically revealed that AmAtg2B interfered with lipophagy and then caused abnormal lipolysis in the pupal stage.

## 1. Introduction

Autophagy is a fundamental cellular process that eliminates molecules and subcellular elements, including nucleic acids, proteins, lipids, and organelles, via lysosome-mediated degradation to promote homeostasis, differentiation, development, and survival [1,2]. Since the discovery of the highly conserved autophagic machinery, numerous autophagy-related genes (Atgs) have emerged [3]. Although autophagy is a highly dynamic, multi-step process, it has a central event, the formation of an autophagosome membrane. Atg8/microtubule-associated protein 1A/1B light chain 3 beta, LC3 (LC3-I and LC-II), is the main molecular marker of autophagosome formation [2]. Autophagy-related protein 2 (Atg2), which has a lipid transfer capacity, is involved in the central event during autophagy; this has been demonstrated in both mammals and yeast [4,5]. The autophagosome is delivered to, and fuses with, the lysosome or vacuole whose contents are subsequently degraded [4]. P62/SQSTM1 bound poly ubiquitinated proteins become incorporated into the completed autophagosome and are degraded in autolysosomes, thus serving as an index of autophagic degradation [6,7]. LC3 and P62 are marker proteins that are successfully used to assay autophagic activity in insects, mammals, zebrafish, and yeasts [4,8,9,10,11].

Autophagy can be divided into selective autophagy and nonselective autophagy [12]. Selective autophagy can be divided into many subtypes [various macromolecules (glycophagy and lipophagy), mitochondria (mitophagy), the endoplasmic reticulum (ER) (ER-phagy), parts of the nucleus (nucleophagy), pathogens (xenophagy), and lysosomes themselves (lysophagy)] based on the specific cargos involved [13]. Lipophagy is a type of selective autophagy owing to the specific autophagic degradation of lipid droplets (LDs) [14]. As triacylglycerol (TAG) stored in LDs is degraded in the lysosome by acid lipase (LAL), TAG breakdown caused by lipophagy is often called “acid” lipolysis [15]. However, there is also a classical pathway for TAG mobilization in LDs, which is achieved through an enzymatic process involving LD-associated lipases (adipose triglyceride lipase (ATGL), hormone-sensitive lipase (HSL), and monoglyceride lipase) [16,17,18]. The mechanisms are different from lipophagy, and TAG breakdown by lipolysis at a pH close to 7.0 is often called “neutral” lipolysis [15]. It has been hypothesized that the two lipolysis processes coexist in organisms because of the difference in the amount of TAG breakdown at the same time. In the case of lipolysis, more TAG breakdown in a short time creates more free fatty acid (FFA) for β-oxidization [19].

Lipophagy was first clearly defined in mouse hepatocytes [14]. Subsequently, it was identified in hypothalamic agouti-related peptide (AgRP) neurons [20], macrophage foam cells [21], enterocytes [22], and adipocytes [23] and contributes greatly to fat mobilization. In addition, lipophagy has been discovered in *Fusarium graminearum* [24], rice [25], *C. elegans* [26,27], yeast [28,29], and zebrafish [10]. In insects, the fat body is the main organ of lipid storage and metabolism [30] and LDs in the fat body are targets of autophagic destruction during lipophagy [31]. Autophagy in the fat body cells of larvae was first recognized in *Drosophila melanogaster* [32,33] and *Rhodnius* [15,34]. More direct evidence suggests that autophagy takes place in the fat body to provide fuel for metamorphic remodeling during the non-feeding pupal stage in *Drosophila melanogaster* [35,36,37,38,39]. Honey bees complete their life cycle by metamorphosis. In the pupal stage, they do not eat any more, and provide energy for pupal development by the nutrients stored in fat body in the early stage [40,41]. LDs are abundant in vegetative cells of fat body. As special organelles in vegetative cells [42], LDs are not only serve as passive lipid storage, but also play a key role in energy metabolism [43]. Therefore, lipolysis and lipophagy are most likely to coexist in the pupa stage of honey bee metamorphosis, which makes honey bees an ideal model for studying the mechanisms between lipophagy and lipolysis. 

LDs are targets of autophagic destruction in lipophagy, but it has also recently been shown that LDs contribute to autophagosome formation in human tumor cells [44]. The Atg2 (Atg2A and Atg2B), Atg14L, and LC3 have been identified as localizing and regulating both LDs and autophagosome biogenesis [44,45,46], which suggests that Atg proteins are essential for both the biogenesis of LDs and autophagosomes, either by fulfilling two independent roles, or perhaps by contributing to regulate lipid transfer from LDs to the site of autophagosome formation [44]. Several years of intense investigation uncovered that Atg2 plays an indispensable role in lipid transport and formation of the autophagosome membrane in yeast and mammals [4,5,47]. At present, the function of Atg2 in autophagy has been systematically performed in yeast and mammals [3,4,5,48], but it has not been studied in insects, and it is not clear whether Atg2 is involved in the regulation of honey bee pupal lipid metabolism and energy supply. 

In this study, it was found that lipophagy occurred in the pupal fat body where there was a specific high expression of *A. mellifera* autophagy-related protein 2 homolog B (*AmAtg2B)*. At the level of translation, we explored the role of AmAtg2B in lipid metabolism and related activities of honey bee pupae by the neutralizing antibody of AmAtg2B. Then, the differential lipid metabolites were screened by LC/MS non-targeted lipidome, and PC and C17 were used to successfully rescue the pupae that had been arrested due to the neutralizing antibody of AmAtg2B. At the transcriptional level, we used RNA interference (RNAi) to investigate the function of AmAtg2B in lipid metabolism and related activities of honey bee pupae. Finally, we confirmed that the AmAtg2B participates in lipophagy and interferes with pupal lipolysis, which is essential to lipid metabolism and energy supply in the pupal stage. Our findings regarding lipophagy in lipid lipolysis expands the functional role of autophagy in the regulation of physiological activities by highlighting its contribution to the maintenance of energy dynamic balance, and provide a new approach for the prevention and control of insects.

## 2. Results

### 2.1. Lipophagy in the Fat Body of A. mellifera Pupae

The relative mRNA expression levels of autophagy indicator protein *LC3* and *AmAtg2B* in Pp, Pw, and Pb were measured. The results showed that the relative mRNA expression levels of the two genes in the pupal stage were significantly higher than those in the PP, which indicated that the latter stage was an active stage of autophagy (Figure 1a). As fasting is needed prior to larval transformation into pupae, starvation is the most likely cause of autophagy [49]. Therefore, the relative mRNA expression levels of starvation-induced gluconeogenetic genes [*fructose-1,6-bisphosphatase* (*fbp*) and *Phosphoenolpyruvate carboxykinase 1* (*pepck1*)] [50,51] were validated by qRT-PCR. The upregulations of *fbp* and *pepck1* verified the starved-like status of pupae, which suggested that autophagy activity may be induced by starvation in the pupal stage (Figure 1a). More precisely, we demonstrated LDs/LC3 co-localization in the fat body cells by using double immunofluorescence analyses, thus indicating a direct association between LDs and autophagosomes (Figure 1b).

### 2.2. Autophagy Is Functionally Involved in the Regulation of Pupal Lipid Metabolism in A. mellifera

Since lipophagy occurred in pupae, we set out to investigate whether whole body inhibition of autophagy would have any effect on lipid metabolism. At first, autophagy can be effectively inhibited (higher expression of P62 and lower expression of LC3-II) after the autophagy inhibitors (3-MA, CQ) are injected into the hemolymph of honey bees (Figure 2a). The energy supply from TAG in the fat body of insects is the main source of energy in the pupal stage [30,52], therefore we measured the content of TAG in the body and the activities of enzymes related to fatty acid (FA) decomposition (ATGL, CPT1) and FA synthesis (ACC) in groups 3-MA, CQ, and CK (Figure 2b). As expected, with the development of honey bee metamorphosis (PP, Pw, Pb), TAG content gradually decreased and remained in a state of consumption (Figure 2c). However, abnormal TAG accumulation occurred in group 3-MA and CQ of Pb owing to inhibited autophagy, and even though the enzyme activities of ATGL, CPT1, and ACC were significantly increased over those in CK (Figure 2c). We conducted non-targeted lipidomics to determine the global differential lipid metabolites between group 3-MA and CK, and the results showed that in addition to the high abundance of TAG and ceramide (Cer), glycerol phospholipids were mainly significantly reduced (Figure 2d). In addition, it has been demonstrated that glucose-sensitive neuropeptide F (NPF) can regulate lipid metabolism through glucagon-like and insulin-like hormones in *Drosophila melanogaster* [51]. Also, as has recently been demonstrated, the honey bee amino acid-sensitive receptor AmGr10, which is activated with L-amino acids and regulates lipid metabolism in the fat body of honey bees through the target of rapamycin (TOR) and the insulin/insulin-like signaling (IIS) pathways [53]. To assess whether inhibited autophagy is an effector of abnormal lipid metabolism in pupae, we examined the relative expression levels of genes involved in lipid lipolysis (*lipase 1*, *lipase 3*, *HSL*) and related activities [*sugar transporter1* (*sut1*), *Glucose transporter* (*Glut*), *neuropeptide F* (*NPF*), *Bursicon*-*α* (*Burs*), *adipokinetic hormone* (*AKH*), *adipokinetic hormone receptor* (*AKHR*), *ATGL*, *transcription factor Forkhead box sub-group O* (*FOXO*), *insulin receptor* (*InR*)]. The results showed that the expression of almost all genes (except *FOXO*) was decreased after treatment with autophagy inhibitors (3-MA and CQ) compared with CK (Figure 2e). As expected, principal component analysis showed clear separation between the lipid metabolism-related genes of pupae in group 3-MA, CQ, and CK (Figure 2f). 

### 2.3. Tissue Specificity of AmAtg2B and Structural and Functional Prediction of Its Protein

In yeast and mammals, LDs in fat stores are not only the target of autophagy (lipophagy) but also the source of autophagosome membrane formation, in which Atg2 plays an important regulatory role [5,44,54] (Figure 3a). In *A. mellifera*, AmAtg2B transcripts were found to be expressed predominantly in the fat body among six different tissues of Pw (head, thorax, leg, epicuticle, fat body, and gut; Figure 3b). To further explore this function, based on the amino acid sequence of AmAtg2B in *A. mellifera* (NCBI Reference Sequence: XP_026296805.1), a phylogenetic tree was constructed to investigate the evolutionary relationships among AmAtg2B in *A. mellifera* and its homologs in humans, mice, *Caenorhabditis elegans*, and *Saccharomyces cerevisiae*. As shown in Figure 3c, AmAtg2B showed a more closed relationship with humans and mice. In addition, the AmAtg2B protein function domain found in humans and mice also exists in *A. mellifera* in view of MEME domain analysis (Figure 3c). Then, the protein structure encoded by AmAtg2B was analyzed. AmAtg2B is very large (encodes of 2085 amino acids) and contains short stretches of ~100 aa at the N and C termini called chorein domains, which are also found within the VPS13 family [5,44,55]; VPS13 as a lipid transport protein has functions to mediate glycerophospholipid transport between organelles at membrane contact sites [56] (Figure 3d). Amino acid sequence analysis showed that the conserved amino acid residues of AmAtg2B in *A. mellifera* are enriched in both the N- and C-terminal regions (Figure 3e). Figure 3f shows a three-dimensional structure prediction (designed by POLYVIEW-3D) including secondary structures, such as α-helix and β-folding (shown in pink) as the protein domain (N-terminal region of chorein or VPS13) using Pfam. 

### 2.4. AmAtg2B Mediated Lipophagy Disorder Could Cause Abnormal Lipid Metabolism and Finally Lead to Abnormal Development of Pupae in A. mellifera

To ascertain the function of AmAtg2B in pupae, we conducted the Anti-Atg2B antibody neutralization test, which significantly reduced the protein expression level of AmAtg2B in pupae (Figure 4a,b). More importantly, Anti-Atg2B antibody treatment resulted in developmental arrest at the Pw stage, compared with group Rabbit-IgG and CK (Figure 4c). According to our prediction of the function of AmAtg2B above, the low expression of AmAtg2B may weaken the pupal lipophagy activity of honeybees, disturb the energy supply balance, and lead to abnormal development of pupae.

Double immunofluorescence staining of LDs/LC3 co-localization was used to test our speculations. The fluorescence images showed that the fluorescence intensity of LC3 (red) in group Anti-AmAtg2B was significantly weaker than that in the other two groups, and the staining area of the fat body (green) was significantly larger than that in the other two groups (Figure 4d), indicating that lipophagy activity of the fat body was weakened after the treatment of AmAtg2B antibody neutralization. In addition, more direct evidence was provided by non-targeted lipidomics, which screened the differential lipid profiles between group Anti-Atg2B and CK. As shown in differential lipid profile heat maps (Figure 4e), after injecting Anti-Atg2B antibodies into pupal hemolymph, the expression abundance of TG/TAG and DG/DAG was significantly higher than that of the CK group, which was consistent with the results after treatment with the autophagy inhibitors (3-MA or CQ). Moreover, abundance of the three most abundant phospholipids [phosphatidylcholine (PC), phosphatidylethanolamine (PE), and phosphatidylserine (PS)] accounted for 72.5% (29/41) of the differential metabolites with significantly lower abundance. 

### 2.5. Molecular Mechanism of AmAtg2B Involved Lipophagy Affecting Pupal Lipid Lipolysis in A. mellifera

The interplay between lipophagy and lipid metabolism is complex [57], therefore we explored the molecular mechanism of AmAtg2B involved lipophagy affecting lipid metabolism from both protein and transcription levels. As a process of selective autophagy, lipophagy mainly occurs in lipid droplets of the fat body [31,58]. Although the AmAtg2B antibody neutralization test can cause a systemic response, the target organ may only be the fat body. To address the molecular mechanism of AmAtg2B affecting lipid metabolism, we performed LC-MS/MS protein identification to analyze the whole body or LDs proteins between group IgG and Anti-Atg2B. KEGG enrichment analysis was performed after comparison analysis of the identified proteins. The results showed that the differential proteins of both whole-body proteins and LD proteins were enriched in metabolic pathways, only the types of metabolic pathways were different. In summary, compared with the group IgG, the proteins undetected in group Anti-Atg2B were significantly enriched in fatty acid degradation (Figure 5aIV) and in the citrate cycle (TCA cycle) pathway (Figure 5bIV), which were closely related to energy supply in vivo. Therefore, it is reasonable for us to believe that the presence of the Anti-Atg2B antibody in the hemolymph will affect lipid metabolism and energy supply in vivo, leading to abnormal development. RNAi-mediated gene knock-down is a useful tool for assessing gene function in honeybees [59]. RNAi can be induced by feeding or injecting double stranded RNA (dsRNA) to trigger degradation of other RNAs with similar sequences, and this has been used to knock-down the expression of bee genes [20]. Therefore, RNAi was performed to study the role of AmAtg2B in lipolysis. qPCR results showed that dsRNA-AmAtg2B could significantly knock-down the expression of AmAtg2B within 24 h after injecting (Appendix A). In subsequent tests, 12 h after dsRNA-AmAtg2B injection, honeybees were selected for functional verification. qPCR results showed that knock-down of AmAtg2B resulted in the significantly down-regulated expression of autophagy marker gene (*LC3*) and a key gene (*Atg9A*) for lipid droplet mobilization (Figure 6d). On the contrary, the expression of genes related to lipid droplet formation [double-FYVE containing protein 1 (DFCP1) and ras-related protein Rab-18-B (Rab18)], and major lipid droplet structural protein (lipid storage droplets surface-binding protein 2, Lsd2) were significantly increased (Figure 6d). The above results show the close relationship between AmAtg2B and lipophagy, that the consumption of AmAtg2B will block the process of lipophagy, resulting in the accumulation of genes encoding lipid droplet structural proteins. After knock-down of AmAtg2B, the activity of lipophagy was significantly inhibited, and the activity of lipid metabolism also seemed to be suppressed. This manifested as the significantly down-regulated expression of genes (*lipase1* and *lipase3*) involved in lipid lipolysis (Figure 6b), significantly down-regulated expression of genes (*Sut1*, *AKHR*, and *FOXO*) involved in regulating lipid metabolism through glucagon-like hormones, and significantly up-regulated expression of genes (*NPF*, *InR*) involved in regulating lipid metabolism through insulin-like hormones (Figure 6c). That the Anti-Atg2B antibody induced a significant reduction in the three most abundant phospholipids (PC, PE, and PS) prompted us to further investigate the roles of AmAtg2B in PC, PE, and PS syntheses. Among the genes examined, genes involved in the cytidine diphosphate ethanolamine (CDP-Cho) pathway for PC synthesis (Chk and Pcyt1) significantly decreased, and Ptdss involved in PS synthesis also significantly decreased (Figure 6a). 

### 2.6. PC Rescued Abnormal Developmental Arrest of Pupae Induced by Anti-Atg2B

We reasoned that if the shortage of PC is the direct cause of Anti-Atg2B-induced abnormal developmental arrest in pupae, so adding back the PC or C17iso would suppress abnormal developmental arrest. It is worth mentioning that C17iso is reportedly present in total lipids (TL) and to varying degrees in PC, PE, phosphatidylinositol (PI), and PS, with PE having an especially high content [60,61]. Furthermore, the addition of C17iso to lipid can rescue the deficiency of TAG, PC, and PE in *Caenorhabditis elegans* [62]. After the implementation of rescue, some Pw in group Anti-Atg2B turned into Pp, which seemed to break the developmental arrest caused by Anti-Atg2B (Figure 7b1). The statistical results also showed that rescue by PC and C17iso significantly increased the proportion of Pp compared with the Anti-Atg2B group (Figure 7c). Unfortunately, however, there was no improvement in the eclosion rate, and none of the larvae injected with Anti-Atg2B antibody emerged to be adults (Figure 7d). 

## 3. Discussion

As a selective type of autophagy, lipophagy has attracted much attention since its discovery, perhaps because it provides an alternative way for TAG breakdown stored in LDs. Earlier studies provided evidence uncovering a close relationship between lipophagy and lipid metabolism in the liver [23,63], pancreas [64], adipose tissue [23,65], and the hypothalamus [19]. In *Drosophila*, developmentally programmed autophagy and subsequent cell death occur in tissue remodeling during metamorphosis [66,67]. Strikingly, autophagy happens in the fat body to provided fuel for the metamorphic remodeling process during the non-feeding pupal stage, which likely plays important roles during development in *Drosophila melanogaster* [35,36,37,39].

Here, we demonstrated that autophagy occurred in honeybee pupae, which targeted LDs in the fat body as verified by co-localization of LDs with microtubule-associated protein 1A/1B light chain 3 beta (LC3), namely lipophagy; and that the injection of autophagy inhibitors (3-MA or CQ) can cause TAG accumulation and abnormal lipid metabolism in the pupae. The results of the present study shed new light on LD breakdown by lipophagy, supporting the theory that lipophagy plays an indispensable role in the mobilization of nutrient stores to fuel the metamorphic remodeling process during the non-feeding pupal stage of honeybees. Strikingly, lipase activity (ATGL, CPT1, and ACC) in vivo after treatment with autophagy inhibitors (3-MA or CQ) showed that weaker autophagy activity promoted lipase activity in cytoplasm. Similar to a previous study, it indicated that lipophagy contributes directly to the mobilization of lipids from LDs to lysosomes, where luminal lipases mediate their lipolysis, and neutralization of the lysosomal pH would affect the activity of the lysosome-resident lipases without suppressing the activity of cytosolic lipases [20]. Thus, our major challenges were to identify the relative specific contributions of autophagy, lipophagy, and lipolysis to the energy homeostasis of pupae in honeybees, and explore how these pathways influence honeybee development.

Among the numerous ATGs involved in the regulation of autophagy, we chose to explore the function of Atg2 because of its close association with the formation of LDs and autophagosome membranes in mammals and yeast [4,5,47]. In addition, Atg2 is also an important member of the ATGs and has been studied since the early 1990s [3,48]. First, qRT-PCR results showed that AmAtg2B was expressed specifically in the pupal fat body, which indicated that it might participate in the activities of the pupal fat body. Further, bioinformatics analysis showed that AmAtg2B, like autophagy, was highly conserved during evolution and exhibits a close relationship with mammals. It was evident that AmAtg2B may play a similar role in lipid transport between autophagosomes and LDs in honeybees as in mammals. Therefore, it seems likely that AmAtg2B is an important molecule that connects autophagy, lipophagy, and lipolysis of the pupal fat body in *A. mellifera.*

Combined detection work of AmAtg2B transcription levels were further conducted to corroborate this hypothesis. For AmAtg2B antibody neutralization, the low expression 8of AmAtg2B did cause a reduction in lipophagy activity (weaker fluorescence intensity of LC3 (red)), and significantly larger fat bodies. This finding is consistent with that of Wang (2016), who also found that knock-down of both Atg2A and Atg2B resulted in not only defective autophagy, but also in LD aggregation, implying a link between autophagy and lipid lipolysis [31]. The disturbance of lipid metabolism (especially in phospholipid anabolism) by weakened lipophagy after AmAtg2B antibody neutralization in Pw demonstrated the relationship between lipophagy and lipolysis.

In addition, the low transcript level of AmAtg2B also resulted in the down-regulation of *LC3* and Atg9A, which is consistent with previous research that depletion of Atg9A does not only inhibit autophagy but also increases the size and/or number of lipid droplets in human cell lines and *C. elegans*. Moreover, Atg9A depletion blocks the transfer of fatty acids from LDs to mitochondria and, consequently, the utilization of fatty acids in mitochondrial respiration [68]. At the same time, the low transcript level of AmAtg2B also resulted in up-regulated DFCP1, Rab18, and Lsd2, and they were associated with LDs formation [30,69,70]. Finally, the low transcript level of AmAtg2B also resulted in down-regulation of *lipase1* and *lipase3*. Comprehensive analysis showed that AmAtg2B was involved in lipophagy and associated with lipid lipolysis of honeybee pupae. 

FA derived from hydrolysis of TAG stored in LDs can be used to synthesize phospholipids used in phagophore membrane expansion [71,72]. The abundance of the three main phospholipids (PE, PC, and PS) decreased after the AmAtg2B antibody neutralization described above. Consistent with results of AmAtg2B knock-down, the above results showed a consistent trend that genes involved in the cytidine diphosphate ethanolamine (CDP-Cho) pathway for PC synthesis (Chk and Pcyt1) significantly decreased, and that the Ptdss involved in PS synthesis also significantly decreased. However, the most important function of FA produced by lipid metabolism is to supply ATP for development as substrates of mitochondrial β-oxidation [73]. Abnormal lipid metabolism caused by Anti-Atg2B-induced weakened lipophagy will inevitably affect the balance of energy supply, which leads to abnormal developmental arrest of honeybee pupae. 

Moreover, FA can be synthesized endogenously from glucose or proteins, a process known as de novo lipogenesis (DNL) [74,75,76]. In mammals, glucose enters the adipocyte through insulin-sensitive (GLUT4) and non-insulin-sensitive (GLUT1) glucose transporters, and then is metabolized through glycolysis and the TCA cycle to produce citrate molecules that are required for DNL [77]. Recently, a similar study related to glucose and lipid metabolism was reported in *Drosophila melanogaster* [51], which uncovered that the sugar-responsive enteroendocrine neuropeptide F regulates lipid metabolism through glucagon-like and insulin-like hormones. After knock-down of the AmAtg2B, we measured the transcription levels of related genes mentioned in the above study, and the results showed that *Sut1*, *AKHR*, and *FOXO* were significantly down-regulated, and *NPF*, *InR* were significantly up-regulated. These data support that low expression of AmAtg2B in pupae hindered the progress of glucose and stored lipids being mobilized to the TCA cycle to generate energy. 

Protein catabolism also could support DNL by providing branched-chain amino acids (BCAAs) [75]. Moreover, an unexpected link between mitochondrial BCAA catabolism and DNL has been identified [76]. A study in *C. elegans* provided more direct evidence for BCAAs and showed that the lipid biosynthesis pathway [78] and BCAAs can be mobilized during fasting [74]. In view of the above research results, the rescue scheme of BCAAs (C17iso) was also designed after the treatment of AmAtg2B antibody neutralization, and the results showed that rescue by both PC and C17iso significantly increased the proportion of Pp compared with the Anti-Atg2B group. But unfortunately, there was no improvement in the eclosion rate, and none of the larvae injected with Anti-Atg2B antibody emerged to be adults.

Collectively, the available experimental evidence indicates that the low level of AmAtg2B in transcription and translation can reduce the activity of lipophagy, affect lipid lipolysis in vivo, and lead to abnormal developmental arrest (Figure 8). Although our finding provided the evidence that lipophagy is an integral part of lipid metabolism and energy homeostasis, the relationship between lipophagy and lipid metabolism could not completely explain the specific molecular mechanism. Therefore, the next step is to explore the molecular level interactions in pathways and explore how these pathways influence the honey bee activity patterns. At the same time, studies on the timing of lipophagy induction and how autophagosomes recognize LDs are also future research directions. The research regarding lipophagy in lipid lipolysis expands the functional role of autophagy in the regulation of physiological activities and providing a reference for exploring therapeutic targets of human diseases with lipid over-accumulation, such as those that comprise the metabolic syndrome.

## 4. Materials and Methods

### 4.1. Honeybee Samples and Specimens

The colonies of *A. mellifera* used in this experiment were collected from the apiary of Science and Technology Innovation Park of Shandong Agricultural University (Tai’an, Shandong, China). The first instar larvae of workers were reared, and larvae were transferred to 24-well plates covered with paper where they prepared to enter the pupation stage after defecation, in vitro, according to a previously described method [79,80]. Six larvae at the prepupa stage (PP, fifth instar larva), white-eyed pupa (Pw, unpigmented cuticle), and brown-eyed pharate-adult (Pb, unpigmented cuticle), were sampled to detect the expression levels of genes and proteins. Ten white-eyed pupae were dissected in insect saline solution (0.1 M NaCl, 20 mM KH_2_PO_4_, 20 mM Na_2_HPO_4_) [81] using ophthalmic tweezers and scissors, and their fat bodies were collected for subsequent digestion into single cells for immunofluorescence colocalization (LDs and LC3).

In an autophagy inhibition experiment, 0.5 μL 3-methyladenine (3-MA, 10 mM), and chloroquine (CQ, 5 mM) were injected into the hemolymph of larvae at the PP stage. The same amount of phosphate buffered saline (PBS) was injected for controls (CK). 

For neutralization of AmAtg2B in pupae, 4 μg of Atg2B rabbit polyclonal antibody (Anti-Atg2B) was injected into the hemolymph of Pw. Equal amounts of rabbit IgG were injected as negative controls (rabbit-IgG). At the same time, a non-injection group (CK) was set as the control group to indicate the normal development of the honeybee pupal stage. For frozen sections, the abdomens of Pb were fixed in 4% neutral formaldehyde for 4 h, 5 μm slices were cut and stored in −20 °C for subsequent tests.

For the phosphatidylcholine (PC) and the monomethylated branched-chain fatty acid (mmBCFA) 15-methylhexadecanoic acid (C17iso) rescue experiment, approximately 48 h after the first injection (pink-eyed pupa in CK), the Anti-Atg2B group was injected with PC and C17iso, and the remaining groups were injected with the same amount of PBS. The final concentration of PC and C17iso was 5 mg/mL at 10 mM. The implementation details of the experimental items are listed in Appendix A.

### 4.2. RNA Preparation and Fluorescent Real-Time Quantitative PCR (qRT-PCR)

The RNAiso Plus kit (TaKaRa, Dalian, China) was used to extract the total RNA of honey bees according to the explanatory memorandum. Then, single-stranded complementary DNA (cDNA) was synthesized using a Transcript All-in-One First-Strand cDNA Synthesis SuperMix (TransGen Biotech, Beijing, China) at 25 °C for 15 min, 55 °C for 30 min, and then at 85 °C for 5 min. The final products were stored at −20 °C. To measure gene expression, fluorescent qRT-PCR was performed using an SYBR PrimeScript RT-PCR kit (TaKaRa, Dalian, China). Fold changes of genes was calculated using the 2^−ΔΔCt^ method [82]. The amplification of the actin transcript (Gene ID: LOC108003299) was used as a sample control [83]. Primers for genes are listed in Appendix A.

### 4.3. Western Blotting

Five larvae in PP, Pw and Pb were homogenized in 1ml of RIPA buffer with protease inhibitor cocktail (CW2200, CoWin Biosciences, Nanjing, China). After centrifugation at 14,000× *g* for 15 min, the supernatant was collected and then 10 μL of supernatant were used to determine protein concentration by using the Micro BCA Protein Assay Kit (CW2011S, CoWin Biosciences, Nanjing, China). Sample loading buffer and RIPA buffer were used according to the extracted protein concentration to a final concentration of 2 μg/μL and boiled for 10 min. Next, a denatured protein mixture of each sample (40 μg) was electrophoresed through a precast 4–12% polyacrylamide gel (ACE Biotechnology, Nanjing, China). Proteins were transferred to a PVDF membrane (Merk Millipore, Saint Louis, MO, USA), which was blocked with NcmBlot blocking buffer (P30500, New Cell & Molecular Biotech, Suzhou, China), incubated with primary antibody at 4 °C overnight, labeled with secondary antibody, and finally detected using a chemiluminescence method with NcmECL Ultra (P10300, New Cell & Molecular Biotech, Suzhou, China). Primary antibodies used were rabbit anti-LC3 (1:1000; Proteintech: 14600-1-AP), rabbit anti-Atg2B (1:1000; ABclonal: A84998), or mouse anti-α-Tubulin (1:2000; Proteintech: 66031-1-lg). α-Tubulin was used for normalization.

### 4.4. Immunofluorescence Staining and BODIPY 493/503 Staining

For lipid droplet staining, the fat bodies of honeybee pupae were dissected and digested with 0.25% trypsin for 30 min, then filtered with cell filter (20 μm), and digestion was terminated by adding serum. The cell-free filtrates were washed in PBS and fixed in 4% paraformaldehyde at room temperature (RT) for 15 min. After the permeation and blocking steps, the cells were incubated overnight with primary antibody rabbit anti-LC3 (1:300; Proteintech: 14600-1-AP), and with the fluorescent secondary antibody (1:50; Abways: AB0131) for 2 h at RT. For BODIPY 493/503 staining, the BODIPY 493/503 stock solution (1 mg/mL) was prepared in DMSO. Stock solution was diluted to a final concentration of 10 μg/mL and used for staining. The samples were incubated for 2 h at RT, cleaned with PBS, observed, and photographed under a laser confocal microscope (Dragonfly200, Andor, Belfast, capital of Northern Ireland, United Kingdom).

For frozen sections, the sections were fixed with 4% paraformaldehyde for 15 min at RT and washed with PBS-Tween (1%, PBST). After the permeation and blocking steps, the cells were incubated overnight with primary antibody rabbit anti-LC3 (1:300; Proteintech: 14600-1-AP), and with the fluorescent secondary antibody (1:50; Abways: AB0131) for 2 h at RT. For BODIPY 493/503 staining, the BODIPY 493/503 stock solution (1 mg/mL) was prepared in DMSO. Stock solution was diluted to a final concentration of 10 μg/mL and used for staining. The samples were incubated for 2 h at RT, cleaned with PBS, observed, and photographed under a laser confocal microscope (Dragonfly200, Andor, Belfast, capital of Northern Ireland, United Kingdom).

### 4.5. Measurement of Enzymatic Activity Levels and TAG Content

Total protein concentrations were measured using a Micro BCA Protein Assay Kit (CW2011S, CoWin Biosciences, Nanjing, China). The enzymatic activity levels of ATGL, Carnitine lipoyl transferase I (CPT1), and Acetyl-CoA carboxylase (ACC) were detected using an Enzyme Linked Immunosorbent Assay Kit (MEIMIAN, Nanjing, China). The content of TAG was measured using the microplate hybrid capture method (Nanjing Jiancheng Bioengineering Institute, Nanjing, China). Subsequently, all measurements were corrected by calibrating the total protein concentration.

### 4.6. Bioinformatics Analysis

Phylogenetic analyses of AmAtg2B were carried out using Molecular Evolutionary Genetics Analysis (MEGA v.4.1, Mega Limited, Auckland, New Zealand) and the MEME (https://meme-suite.org/meme/tools/meme) (accessed on 7 March 2020). The 3DLigandSite-Ligand binding site prediction Server (http://www.sbg.bio.ic.ac.uk/3dligandsite/) (accessed on 26 March 2020) was used to predict the ligand-binding sites [84] and a three-dimensional structure was obtained with the help of a web-based tool (POLYVIEW-3D) (http://polyview.cchmc.org/polyview3d.html) (accessed on 29 March 2020) [85]. 

### 4.7. LC-MS/MS Protein Identification and LC/MS Non-Targeted Lipidome Analysis

Lipid drops in Pb (group IgG and Anti-Atg2B) were extracted using a Lipid Droplet Isolation Kit (MET-5011, CELL BIOLABS, Shanghai, China). The larval homogenate and extracted lipid droplet protein were denatured and subjected to SDS-PAGE, and the protein gel was stained with Coomassie brilliant blue rapid staining solution (RB010-500, ruibiotech, Beijing, China). Strips of one lane in each group were used for LC-MS/MS protein identification. Six pupae were sampled in each experimental group for LC/MS non-targeted lipidome analysis. The results of LC-MS/MS protein identification and LC/MS non-targeted lipidome analysis were conducted at Oebiotech. 

### 4.8. RNA INTERFERENCE

RNA interference was performed at the white-eyed pupa stage (Pw). Long dsRNAs (>500 bp) can be used in worms [86] and insects [87] and can be broken down into several smaller dsRNA in vivo [88]. The long dsRNA of AmAtg2B (LOC726497) and the dsRNA of green fluorescent protein gene (GFP control; GenBank accession number U87974) were synthesized using RiboMAX T7 large-scale RNA production systems (P1320, Promega, Beijing, China), according to the manufacturer’s instructions. To remove the DNA template, the synthesized dsRNA was digested using DNase I, precipitated with absolute ethyl alcohol, and then redissolved in RNase-free water. Honeybees injected with GFP/water were used as a control [89]. Six honey bees were taken every 12 h for a total of four times over 48 h, labeled, frozen in liquid nitrogen, and immediately stored at −80 °C.

### 4.9. Statistical Analysis

SAS (Version 9.1; SAS Institute Inc., Cary, NC, USA) was used for data analysis and Prism (Version 9.0; GraphPad, La Jolla, CA, USA) was used to create Figures. A Student’s t-test or one-way ANOVA with a post hoc Tukey’s honest significant difference test (Tukey HSD) was used to evaluate statistical significance. Adobe Illustrator 2020, and Adobe Photoshop CS5 were also used for the output of graphic results. https://www.omicshare.com/tools/Home/Soft/getsoft (accessed on 14 June 2022) was used for data analysis. Presented data are means ± SEM and *p*-values < 0.05 were considered statistically significant.

## Figures and Tables

**Figure 1 ijms-24-02096-f001:**
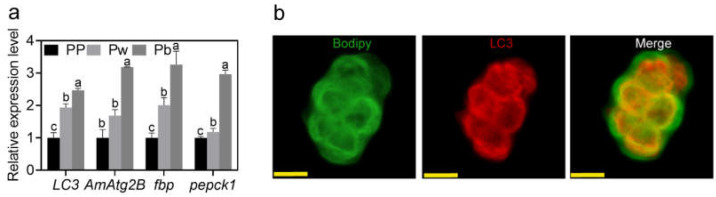
Lipophagy in the fat body of *Apis mellifera* pupae. (**a**) Fold changes of autophagy related genes (*LC3* and *AmAtg2B*) and starvation-induced gluconeogenetic genes (*fructose-1,6-bisphosphatase* (*fbp*) and *Phosphoenolpyruvate carboxykinase* 1 (*pepck1*)) in stage PP, Pw, and Pb. (**b**) Co-localization of LDs (green) with LC3 (red) in fat body cell of Pb. Scale bar: 5 μm. Note: Data are shown as mean ± SEM and the different letters marked on the bar chart represent significant differences (*p* < 0.05).

**Figure 2 ijms-24-02096-f002:**
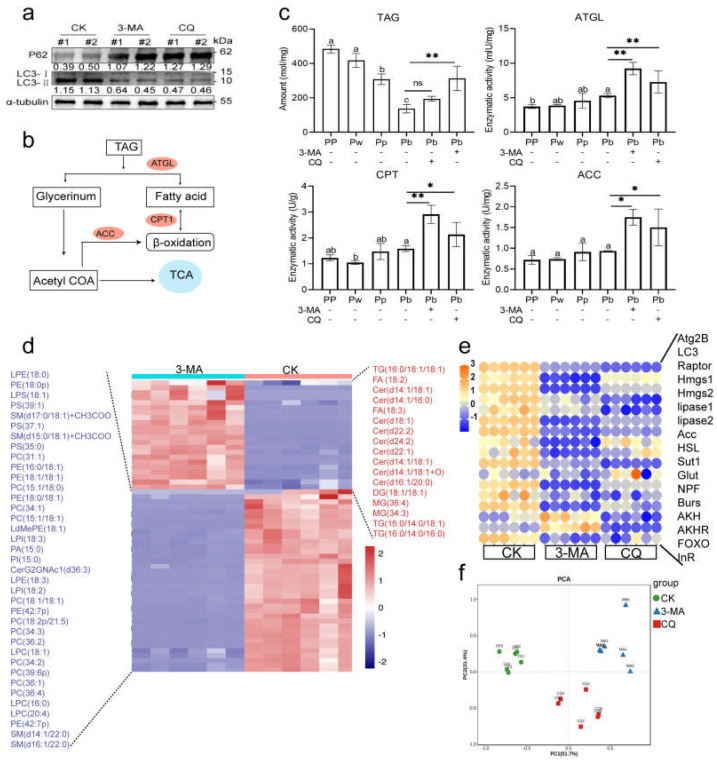
Effects of autophagy inhibitors (3-MA and CQ) on lipid metabolism during pupal stage in *Apis mellifera.* (**a**) Western blot images showing levels of autophagy indicator proteins (LC3 and P62) in Pw with the autophagy inhibitors (3-MA and CQ) treatment. (**b**) Schematic of TG lipolysis for β-oxidation. (**c**) Determination of TAG content and rate-limiting enzymes in TAG lipolysis for β-oxidation both in stage PP, Pw, and Pb and under the treatment of autophagy inhibitors (3-MA and CQ) in Pb. (**d**) Heatmap showing the differential lipid metabolites between group 3-MA and CK. (**e**) qRT-PCR analysis of expression of genes involved in lipid metabolism and related activities. (**f**) Principal component analysis plot of qRT-PCR data in 3-MA, CQ and CK. Note: Data are shown as mean ± SEM and the different letters or * marked on the bar chart represent significant differences (*p* < 0.05), ** (*p* < 0.01), ns, non-significant (*p* > 0.05).

**Figure 3 ijms-24-02096-f003:**
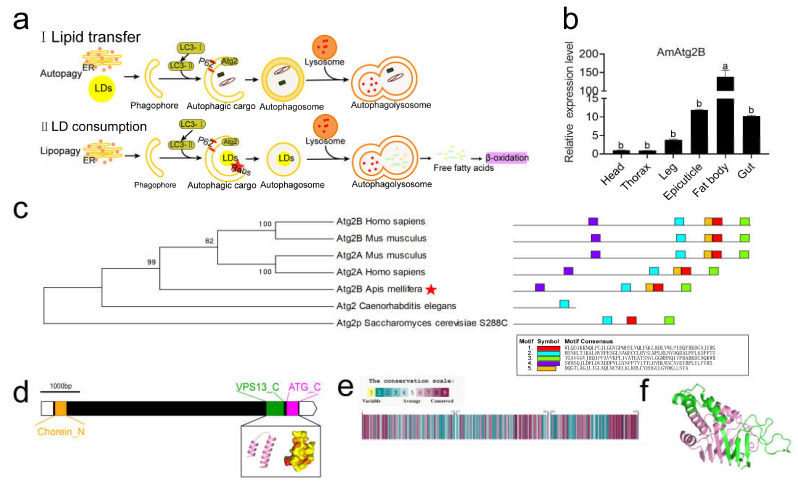
AmAtg2B tissue specificity expression and AmAtg2B protein structure and function analysis. (**a**) The function of AmAtg2B in mammals and yeasts: autophagosome formation. The lipid source of autophagosome membrane may be endoplasmic reticulum (i) or lipid droplets (ii); (**b**) Expression of AmAtg2B in different tissues as detected by qRT-PCR, data are shown as mean ± SEM and the different letters marked on the bar chart represent significant differences (*p* < 0.05); (**c**) Phylogenetic tree analysis of AmAtg2B and analysis of conserved domain in five species (*Apis mellifera*, *Human*, *Mouse*, *Caenorhabditis elegans* and *Saccharomyces cerevisiae*), AmAtg2B was marked with an asterisk; (**d**) protein structure analysis of AmAtg2B, composition of amino acid conserved domain; (**e**) amino acid conserved score; and (**f**) three dimensional structure prediction.

**Figure 4 ijms-24-02096-f004:**
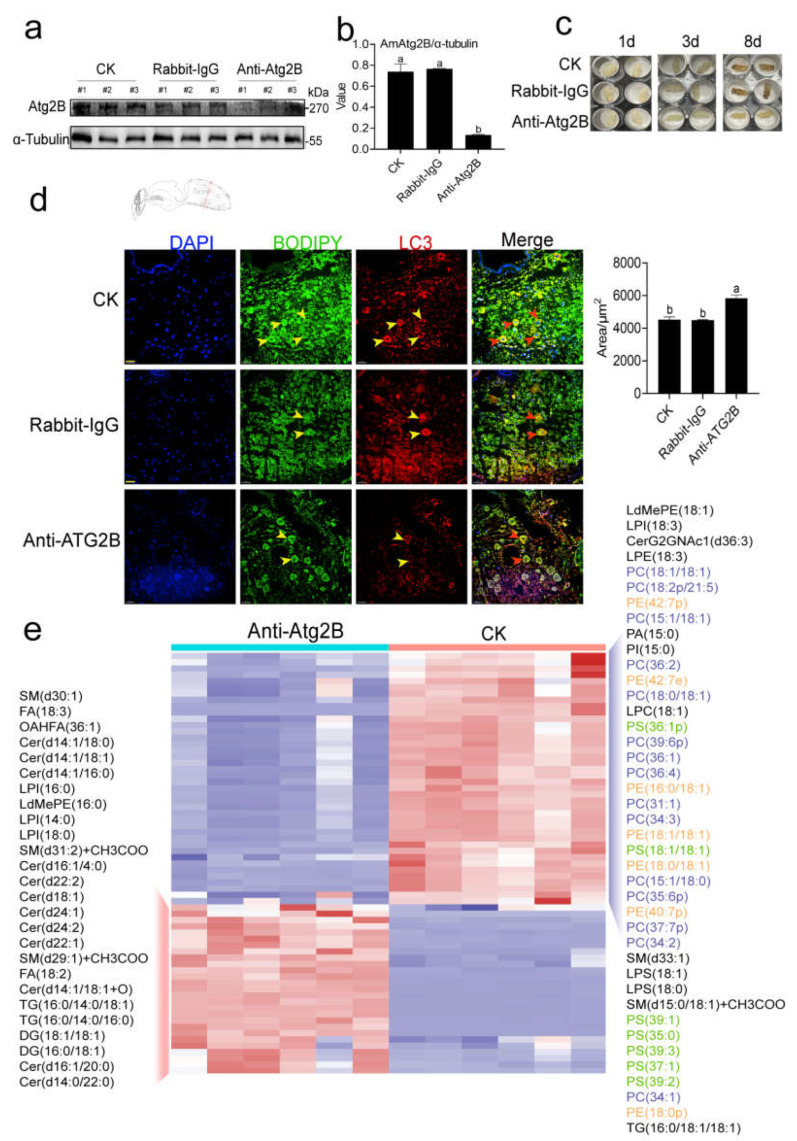
Neutralization of AmAtg2B repressed lipophagy of fat body and lipid metabolism in *Apis mellifera* pupae. (**a**) Western blot image showed that AmAtg2B levels of pupae decreased with the injection of Anti-Atg2B antibodies, compared with group Rabbit-IgG and CK. Rabbit-IgG was a negative control for the antibody and statistical analysis of normalized quantification of mean gray intensity by the ImageJ software (*n* = 3) (**b**). (**c**) Phenotype of pupae after injection of Anti-Atg2B antibodies for 1 d, 3 d, and 8 d (4 μg was injected into Pw); (**d**) Co-localization of BODIPY 493/503 (green) with LC3 (red) in group CK, IgG and (Anti-Atg2B). Statistical analysis of the fat body area (green) by the ImageJ software (*n* = 6). Scale bar: 100 μm. (**e**) Heatmap showing the differential lipid metabolites between group CK and Anti-Atg2B. Note: Data are shown as mean ± SEM and the different letters marked on the bar chart represent significant differences (*p* < 0.05).

**Figure 5 ijms-24-02096-f005:**
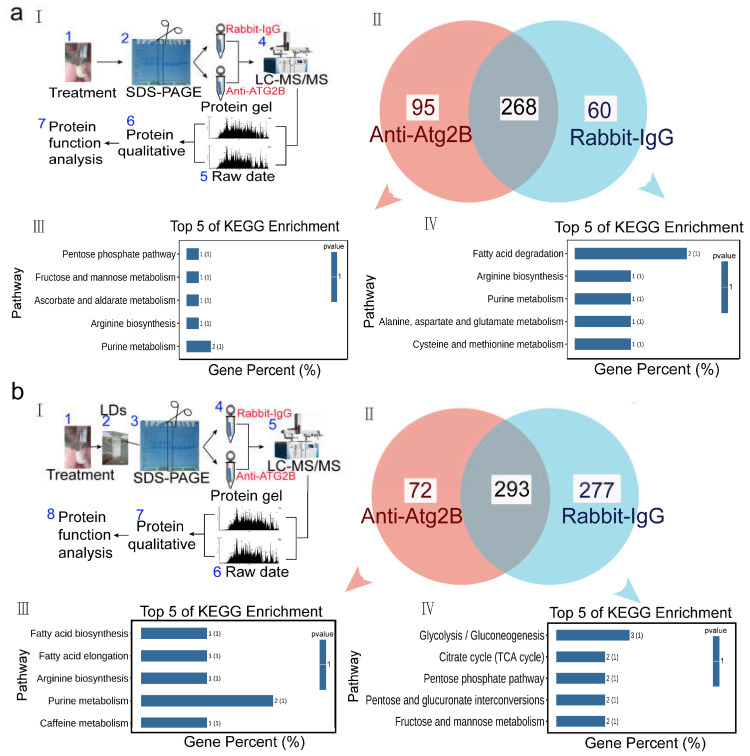
Changes of proteins expression after Anti-Atg2B antibody neutralization of *Apis mellifera* pupae. LC-MS/MS protein identification were conducted to analysis the whole body (**a**) or lipid droplet protein; (**b**) profiles between group IgG and Anti-Atg2B. (**I**) Flow Chart of the LC-MS/MS protein identification; (**II**) Venn diagram analysis of the identified proteins; KEGG enrichment analysis was performed after comparison analysis of the identified proteins in group Anti-Atg2B (**III**) and IgG (**IV**).

**Figure 6 ijms-24-02096-f006:**
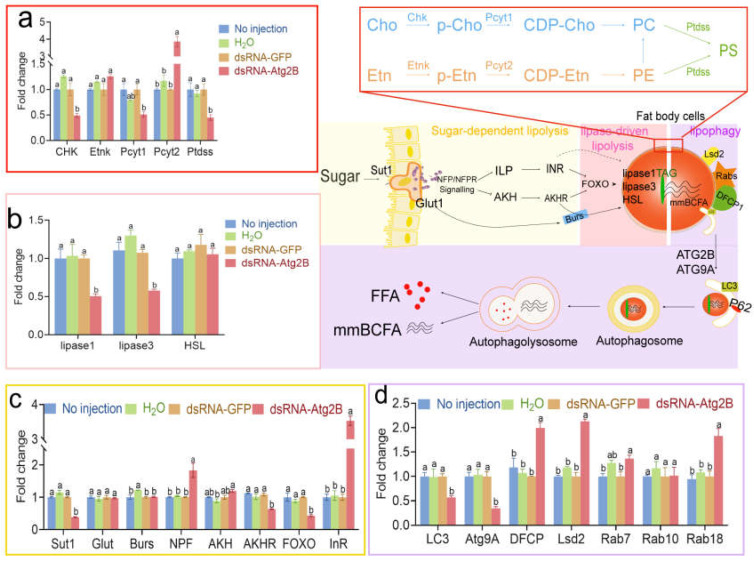
Fold changes of genes related to lipid lipolysis after knockdown the AmAtg2B of pupae in *Apis mellifera*. (**a**) Fold changes of *LC3* and genes for lipid droplet (*ATG9A*, *DFCP1*, *Lsd2*, *Rab7*, *Rab10* and *Rab18*); (**b**) fold changes of genes for traditional lipase-driven lipolysis (*lipase1*, *lipase3* and HSL); (**c**) fold changes of genes for the regulation of sugar-dependent lipolysis (*Sut1*, *Glut*, *Burs*, *NPF*, *AKH*, *AKHR*, *HSL*, *FOXO* and *InR*); (**d**) qRT-PCR analysis of expression of genes encoding rate-limiting enzymes in phospholipid synthesis pathways after knockdown the AmAtg2B of pupae. Note: Data are shown as mean ± SEM and the different letters marked on the bar chart represent significant differences (*p* < 0.05).

**Figure 7 ijms-24-02096-f007:**
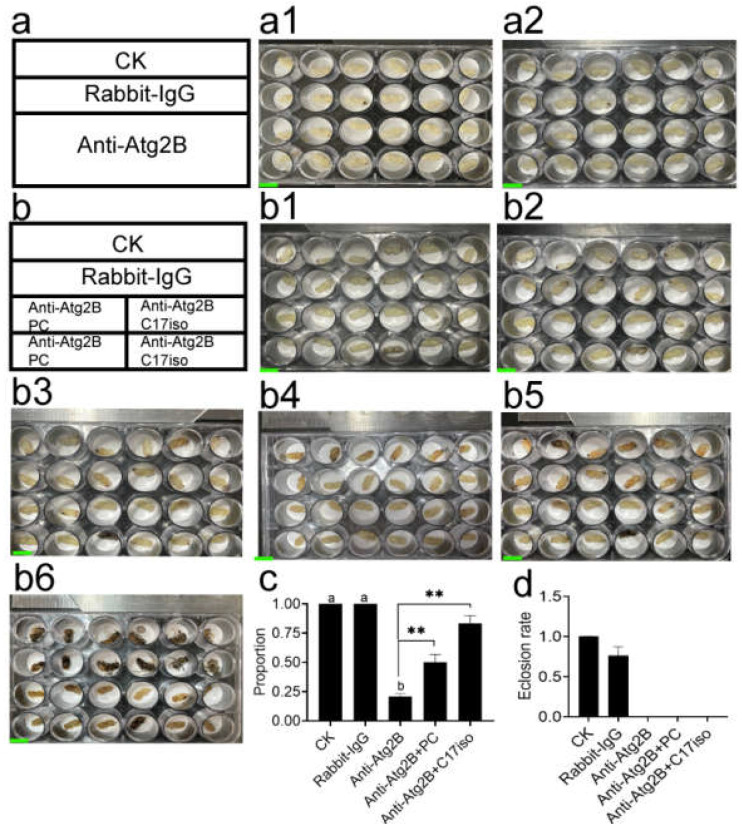
The developmental arrest of Anti-Atg2B-induced pupae is partially rescued by injecting PC or C17iso. (**a**) The first two days (**a1**,**a2**) developmental progression of pupae after the injection of Anti-Atg2B antibodies or IgG and CK. (**b**) Schematic description of PC or C17iso rescuing experiments, and the continuous growth recording until emergence in CK (**b1**–**b6**). Scale bar: 1 cm. (**c**) Growth arrest of Anti-Atg2B-induced pupae could be partially rescued into Pp by injecting PC or C17iso. (**d**) Percentage of eclosion in rescued group, IgG and CK. Note: Data are shown as mean ± SEM and the different letters marked on the bar chart represent significant differences (*p* < 0.05), ** (*p* < 0.01).

**Figure 8 ijms-24-02096-f008:**
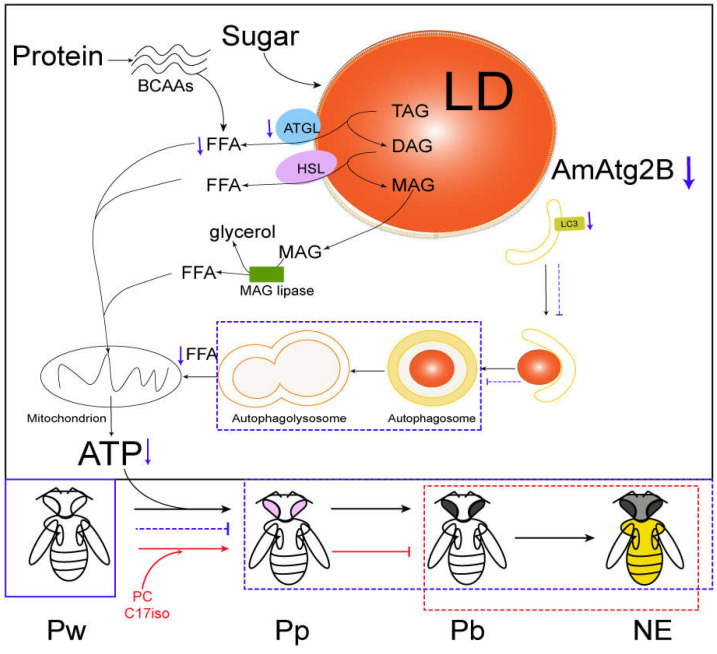
Proposed model of the AmAtg2B function. Low level of AmAtg2B in transcription and translation can reduce the activity of lipophagy, affect lipid lipolysis in vivo, and lead to abnormal development arrest at Pw, which rescued by both PC and C17iso significantly increased the proportion of Pp (Figure 7c). Unfortunately, there was no improvement in the eclosion rate, and not all of the pupae injected Anti-Atg2B antibody could emerged as adults. PP: prepupa, Pw: white-eyed pupae, Pp: pink-eyed pupae, Pb: black-eyed pupae; NE: Newly emerged adult.

## Data Availability

All data are contained within the manuscript, including Appendix A.

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
