# Peer review of "AmAtg2B-Mediated Lipophagy Regulates Lipolysis of Pupae in *Apis mellifera"

_ijms, 2023, doi:10.3390/ijms24032096_

Round 1
Reviewer 1 Report
Reviewer comments
Manuscript ijms-2074699 - AmAtg2B-mediated lipophagy regulates lipolysis of pupae in Apis mellifera
The authors studied lipophagy in honeybee pupae, which targeted lipid droplets in fat body, which was verified by co-localization of lipid droplets with microtubule-associated protein 1A/1B light chain 3 beta (LC3). Moreover, AmAtg2 homolog B (AmAtg2B) was expressed specifically in pupal fat body, which indicated that AmAtg2B might have special function in fat body. Further, AmAtg2B antibody neutralization and AmAtg2B knock-down were undertaken to verify the functions in pupae. Results showed that low expression of AmAtg2B at the protein and transcriptional levels led to lipophagy inhibition, which down-regulated the expression levels of proteins and genes related to lipolysis. Altogether, results in this study systematically revealed that AmAtg2B interfered with lipophagy and then caused abnormal lipolysis in the pupal stage.
The statistical methods are correct.
The English of the text is well written and well readable.
The uniqueness of the text is more than 90% by AntiPlagiarism.NET.
The text contains some misspellings and typos.
There are following comments:
Line 81 - proteinLC3 - insert space - protein LC3.
Line 121 - After the sentence - In addition, it has been demonstrated that glucose-sensitive neuropeptide F (NPF) can regulate lipid metabolism through glucagon-like and insulin-like hormones in Drosophila melanogaster [48]. - add the sentence - Also, it has recently been demonstrated, the honey bee amino acid-sensitive receptor AmGr10, which is activated with L-amino acids and regulates lipid metabolism in the fat body of honey bees through the target of rapamycin (TOR) and the insulin/insulin-like signaling (IIS) pathways (Lim et al., 2019).
Add to the references: Lim, S., Jung, J., Yunusbaev, U., Ilyasov, R.A., Kwon, H.W., 2019. Characterization and its implication of a novel taste receptor detecting nutrients in the honey bee, Apis mellifera. Scientific Reports. 9 (1), 17004. doi: 10.1038/s41598-019-53738-6.
Line 171 - specifcity - should be - specificity.
Line 174 - diferent - should be - different.
Line 202 - phosphatidycholine - should be - phosphatidylcholine.
Line 314 - Disscussion - should be - Discussion
Line 356 seemes - should be - seems.
Line 361 - luorescence - should be - fluorescence.
Line 362 - bodys - should be - bodies.
Line 422 - lipohagy - should be - lipophagy.
Line 422 - abormal - should be - abnormal.
Line 478 - ireland - should be - Ireland.
Please improve the manuscript according to the above comments.

Author Response
Response to Reviewer 1 Comments
Point 1: Line 121 - After the sentence - In addition, it has been demonstrated that glucose-sensitive neuropeptide F (NPF) can regulate lipid metabolism through glucagon-like and insulin-like
hormones in Drosophila melanogaster [48]. - add the sentence - Also, it has recently been demonstrated, the honey bee amino acid-sensitive receptor AmGr10, which is activated with L-amino acids and regulates lipid metabolism in the fat body of honey bees through the target of rapamycin (TOR) and the insulin/insulin-like signaling (IIS) pathways (Lim et al., 2019). Add to the references: Lim, S., Jung, J., Yunusbaev, U., Ilyasov, R.A., Kwon, H.W., 2019. Characterization and its implication of a novel taste receptor detecting nutrients in the honey bee, Apis mellifera. Scientific Reports. 9 (1), 17004. doi: 10.1038/s41598- 019-53738-6
Response 1: Line 121-154: I finished the revision.
Point 2: The text contains some misspellings and typos.
Response 2: I corrected the spelling and formatting errors one by one.
Note:
In addition to the above contents, I have revised the contents of various parts, mainly including the font size of charts, materials and methods, references, and the supplement of missing contents in the introduction and discussion, which are uniformly displayed in the revised manuscript.

Reviewer 2 Report
The study entitled "AmAtg2B-mediated lipophagy regulates lipolysis of pupae in Apis mellifera" by Chen and colleagues, provides an investigation of the functional and physiological role of the AmAtg2 homolog B protein of Apis mellifera using various techniques and proposing a mechanistic framework, with particular relevance to metamorphosis and post-embryonic development.
In my opinion, the study contains relevant and noteworthy information that improves the limited knowledge in insect physiology and biochemistry and paves the way for further applications. However, some important aspects could be improved. Below is a detailed list of minor and major comments and criticisms:
The role of Apis mellifera in the study is unclear. The authors should introduce the model in the introductory paragraph, emphasizing the characteristics of Apis mellifera as a holometabolic organism and arguing the reasons for the choice, could also include important perspectives on colony collapse and physiological knowledge.
The authors has mixed materials and methods, results and discussion, providing details of the protocol, presenting the data and commenting and formulating hypotheses in Section "2. Results". I would suggest changing the title "Results" to "2. Results and discussion" and "3. Mechanistic framework" instead of “Discussion”. Materials and methods information should all be available in Section 4. Materials and methods, distributing the details in the results (e.g., L. 190-192, 240-242, 244-245) is confusing.
The figures are too small, the text is almost impossible to read (e.g. Fig. 5). In Fig. 4d it is impossible to see what the arrowheads indicate. I would suggest rearranging the panel and increasing the font size.
The final paragraph of the introduction should explain the objectives and questions of the study, the authors have summarized the main findings of the manuscript, in my opinion this section should be revised.
Please standardize the writing of AmAtg2B in the text, which is sometimes written in italics.
I would suggest replacing key words already in the title: AmAtg2B; lipophagy; lipolysis; with bees, development, fat bodies.
In the main text some worlds are underlined as hyperlinked (L. 16, 27, 28, 76), please revise them.
Section 4.1 Honeybee samples and specimens in Materials and Methods is a bit confusing and difficult to follow, a table summarizing the experimental groups and sample size would be helpful.
The authors should explain what the statement at L.436 "Pw larvae" means, since Pw represents the "white-eyed pupa (Pw, unpigmented cuticle)" group and not the larval stage.
L. 443 what does the term nymphs indicate? It seems hardly correct for a holometabolous organism as hymenoptera are.
L.453 The author should explain the meaning of C17iso.
I would suggest moving Table 1 of PCR primers in this study to the supplementary material, including a more detailed description of the protocol used in 4.2 and 4.7.
L. 472 "10 μl of supernatant was used" replace "was" with "were."
Specify how the fatty bodies were filtered after digestion with 0.25% trypsin (L. 488).
L.538 authors should add information on how many samples were stored and how many time points were analyzed.
L. 192 remove results
L. 268 correct a report
The first paragraph of section 3. Discussion (L. 315-329) seems redundant and unrelated to the discussion of the data. I would suggest integrating it into Section 1. Introduction, shortening it.
Figure references are no longer needed in the discussion section (e.g., L. 334, 339, 354, 367, 369), with the only exception of Fig. 7.
I would add future perspectives and impacts of this study at the end of the discussion, emphasizing the physiological and ecological relevance of the finding and also the potential application as a marker in different fields.
Author Response
Response to Reviewer 2 Comments
Point 1: The role of Apis mellifera in the study is unclear. The authors should introduce the model in the introductory paragraph, emphasizing the characteristics of Apis mellifera as a holometabolic organism and arguing the reasons for the choice,could also include important perspectives on colony collapse and physiological knowledge.
Response 1: Line 66-72: I have made corresponding supplements to your comments.
Point 2: The authors has mixed materials and methods, results and discussion, providing details of the protocol, presenting the data and commenting and formulating hypotheses in Section "2.Results". I would suggest changing the title "Results" to "2. Results and discussion" and "3. Mechanistic framework" instead of “Discussion”. Materials and methods information should all be
available in Section 4. Materials and methods, distributing the details in the results (e.g., L. 190-192, 240-242, 244-245) is confusing.
Response 2: Thank you for making me aware of the confusion in the article. Although I kept the original framework results of the article, I modified all parts of the content. I want the status of the article to be concise and the content to be appropriate.
Point 3: The figures are too small, the text is almost impossible to read (e.g. Fig. 5). In Fig. 4d it is impossible to see what the arrowheads indicate. I would suggest rearranging the panel and increasing the font size.
Response 3: I reformatted the chart to increase the size of the text.
Point 4: The final paragraph of the introduction should explain the objectives and questions of the study, the authors have summarized the main findings of the manuscript, in my opinion this section should be revised.
Response 4: Line 92-105: I supplemented the objectives and questions of the study.
Point 5: Please standardize the writing of AmAtg2B in the text, which is sometimes written in italics.
Response 5: To distinguish between proteins and genes, AmAtg2B in the text is the protein and AmAtg2B in the text is the gene.
Point 6: I would suggest replacing key words already in the title: AmAtg2B; lipophagy; lipolysis; with bees, development, fat bodies.
Response 6: Line 21: I replaced the content.
Point 7: In the main text some worlds are underlined as hyperlinked (L.16, 27, 28, 76), please revise them.
Response 7: I canceled the hyperlink.
Point 8: Section 4.1 Honeybee samples and specimens in Materials and Methods is a bit confusing and difficult to follow, a table summarizing the experimental groups and sample size would be
helpful.
Response 8: Table 1. The implementation details of the experimental items.
Point 9: The authors should explain what the statement at L.436 "Pw larvae" means, since Pw represents the "white-eyed pupa (Pw, unpigmented cuticle)" group and not the larval stage.
Response 9: Line 497: "Pw larvae" changed with "white-eyed pupae "
Point 10: L. 443 what does the term nymphs indicate? It seems hardly correct for a holometabolous organism as hymenoptera are.
Response 10: This part contained "nymphs" has been deleted.
Point 11: L.453 The author should explain the meaning of C17iso.
Response 11: Line 514-515: The meaning of C17iso was explained with the monomethylated branched-chain fatty acid (mmBCFA) 15-methylhexadecanoic acid (C17iso).
Point 12: I would suggest moving Table 1 of PCR primers in this study to the supplementary material, including a more detailed description of the protocol used in 4.2 and 4.7.
Response 12: Table 1 of PCR primers in this study and the protocol used in 4.2 and 4.7 would be resubmitted as supplementary material.
Point 13: L. 472 "10 μl of supernatant was used" replace "was" with "were."
Response 13: Line 540: I replaced "was" with "were."
Point 14: Specify how the fatty bodies were filtered after digestion with 0.25% trypsin (L. 488).
Response 14: Line 556: The fatty bodies were filtered with cell filter (20μm) after digestion with 0.25% trypsin.
Point 15: L.538 authors should add information on how many samples were stored and how many time points were analyzed.
Response 15: Line 607-609: I added information on how many samples were stored and how many time points were analyzed.
Point 16: L. 192 remove results
- 268 correct a report
Response 16: I finished them.
Point 17: L. 192 remove results
- 268 correct a report
Response 17: Line 237: I finished them.
Point 18: The first paragraph of section 3. Discussion (L. 315-329) seems redundant and unrelated to the discussion of the data. I would suggest integrating it into Section 1. Introduction, shortening it.
Figure references are no longer needed in the discussion section (e.g., L. 334, 339, 354, 367, 369), with the only exception of Fig.7
Response 18: I accepted all of your suggestions, but I kept Fig. 8 instead of Fig. 7.
Point 19: I would add future perspectives and impacts of this study at the end of the discussion, emphasizing the physiological and ecological relevance of the finding and also the potential application as a marker in different fields.
Response 19: Line 466-476: I accepted all of your suggestions, and supplemented the corresponding content.
Note:
In addition to the above contents, I have revised the contents of various parts, mainly including the font size of charts, materials and methods, references, and the supplement of missing contents in the introduction and discussion, which are uniformly displayed in the revised manuscript.

Round 2
Reviewer 2 Report
The authors have addressed most of my criticisms. however, some minor revisions are still needed; see the detailed comments below:
Point 3: in my opinion the figures are still too small and difficult to read; I suggest at least increasing the size of the figures on the page.
Point 8: the new table 1 is useful for understanding the design, but is too detailed for the main text; I would suggest moving it to the supplementary materials.
Point 12: my previous suggestion was to extend the details of 4.2 and 4.7 in the supplementary material, moving the table of PCR primers there, but keeping a short description in the main text. I would suggest reintroducing the sections as they were in the previous version, moving the table and extending the materials and methods in the supplementary material.
Author Response
Point 3: in my opinion the figures are still too small and difficult to read; I suggest at least increasing the size of the figures on the page.
Response 3: I reformatted the chart to increase the size of the text again and each picture may be clearly presented and readable.
Point 8: the new table 1 is useful for understanding the design, but is too detailed for the main text; I would suggest moving it to the supplementary materials.
Response 8: The new Table 1 would be resubmitted as supplementary material.
Point 12: my previous suggestion was to extend the details of 4.2 and 4.7 in the supplementary material, moving the table of PCR primers there, but keeping a short description in the main text. I would suggest reintroducing the sections as they were in the previous version, moving the table and extending the materials and methods in the supplementary material.
Response 12: I reintroducd the sections 4.2 and 4.7 as they were in the previous version, and extended the materials and methods in the supplementary material.
